# Common Regulatory Mechanisms Mediated by Cuproptosis Genes in Inflammatory Bowel Disease and Major Depressive Disorder

**DOI:** 10.3390/genes16030339

**Published:** 2025-03-14

**Authors:** Jiyuan Shi, Qianyi Wu, Mengmeng Sang, Liming Mao

**Affiliations:** 1Department of Immunology, School of Medicine, Nantong University, 19 Qixiu Road, Nantong 226001, China; 2331310021@stmail.ntu.edu.cn (J.S.); 2331310018@stmail.ntu.edu.cn (Q.W.); 2Basic Medical Research Center, School of Medicine, Nantong University, Nantong 226019, China

**Keywords:** cuproptosis, inflammatory bowel disease, major depressive disorder, machine learning algorithms, molecular docking and dynamics analysis

## Abstract

Background: The prevalence of major depressive disorder (MDD) among patients with inflammatory bowel disease (IBD) is significantly higher compared to the general population, suggesting a potential link between their pathogeneses. Cuproptosis, defined as cell death caused by intracellular copper accumulation, has not been thoroughly investigated in the context of IBD and MDD. This study aims to uncover the molecular mechanisms of cuproptosis-related genes (CRGs) in both conditions and to explore novel therapeutic strategies by the modulation of CRGs. Methods: In this study, we identified differentially expressed CRGs between normal and disease samples. We calculated the correlation among CRGs and between CRGs and immune cell infiltrations across various tissues. Four machine learning algorithms were employed to identify key CRGs associated with IBD and MDD. Additionally, drug sensitivity, molecular docking, and molecular dynamics simulations were conducted to predict therapeutic drugs for IBD and MDD. Results: We identified *DLD*, *DLAT*, *DLST*, *PDHB*, and *DBT* as common DE-CRGs, and *DLD*, *LIAS*, *SLC31A1*, *SCO2*, and *CDKN2A* as key CRGs associated with both IBD and MDD. Consequently, *DLD* was recognized as a shared biomarker in both diseases. A total of 37 potential therapeutic drugs were identified for IBD and MDD. Based on the molecular docking and molecular dynamics simulation analyses, barasertib and NTP-TAE684, which target DLAT, were predicted to be the most effective compounds. Conclusions: These findings have substantially enhanced our understanding of the similarities and differences in the regulatory mechanisms of CRGs within brain–gut axis diseases. Key biomarkers have been identified, and potential therapeutic drugs have been predicted to effectively target IBD and MDD.

## 1. Introduction

Inflammatory bowel disease (IBD), primarily encompassing Crohn’s disease (CD) and ulcerative colitis (UC), represents a group of chronic inflammatory disorders affecting the gastrointestinal tract. These conditions impact individuals across all age groups, leading to substantial morbidity and a significant reduction in quality of life [1]. Over the past decade, the prevalence of IBD has risen from 0.3% to 1.3%, and this trend continues [2]. Common symptoms experienced by IBD patients include weight loss, diarrhea, rectal bleeding, and abdominal pain [3]. Patients with IBD may experience symptoms of common mental disorders such as anxiety and depression, due to bidirectional communication via the gut–brain axis, chronicity of symptoms, impaired quality of life, and reduced social functioning [4]. These symptoms may potentially progress to major depressive disorder (MDD), a condition characterized by somatic, emotional, cognitive, and behavioral conditions [5,6]. Anxiety and depression can be exacerbated by exposure to stressors such as immobilization, social defeat, forced swimming, and pathogen infection [6,7]. Exposure to stressors such as immobilization stress (IS) and social defeat stimulates the secretion of adrenaline and glucocorticoids via the hypothalamic–pituitary–adrenal (HPA) axis, as well as tumor necrosis factor (TNF)-α and interleukin (IL)-6 in immune cells, leading to the onset of anxiety/depression and gut inflammation [8]. The excessive stimulation of glucocorticoids and proinflammatory cytokines causes gut inflammation and sysbiosis by activating innate and adaptive immunities in the gastrointestinal tract [9]. Prolonged IBD increases the chances of developing MDD. Gut microbiota perturbation in IBD patients disrupts the immune and central nervous systems through microbiota–gut interaction and gut–brain communication, contributing to the development of anxiety and depression [10,11]. Moreover, gut microbiota perturbation has been implicated in various intestinal processes, including neoplastic pathology and post-anastomotic leakage. The literature significantly supports the potential of fecal microbiota transplantation as a therapeutic option for colorectal cancer, demonstrating its efficacy in gastrointestinal disorders such as *Clostridioides difficile* infection, UC, and irritable bowel syndrome [12]. The intestinal microbiota appears to play an important role in anastomotic leakage following colorectal resection. Targeted interventions aimed at modulating the composition of the microbiota and addressing the pathophysiological mechanisms that impair anastomotic healing could potentially diminish the risk of anastomotic leakage and improve clinical outcomes [13]. Pathophysiologically, MDD-associated stress has been demonstrated to negatively impact gastrointestinal function and increase gut permeability by affecting the immune, endocrine, and nervous system [14], thereby promoting the relapse of gut inflammation after remission. Of note, a study by Spina et al. [15] revealed that anxious–depressive disorders can affect patients with IBD even in conditions of complete clinical remission.

Cuproptosis is a distinct form of cell death characterized by its copper dependency, setting it apart from other established modes of cell death [16]. Copper is recognized as an essential micronutrient with recommended daily intake for individuals with IBD. As a cofactor of a wide range of enzymes involved in metabolic pathways, copper levels are tightly regulated through concentration gradients, thus preventing intracellular cytotoxicity caused by accumulation [17]. Głąbska et al. demonstrated no significant difference in copper intake between UC patients and healthy individuals [18]. Despite a significant decrease in the serum level of copper among children with CD compared to healthy children [19], no difference was observed in the blood copper concentration between adult UC patients and healthy subjects [20]. As indicated by studies on Wilson’s disease [21,22], a rare autosomal recessive disorder characterized by impaired hepatic copper transport and reduced biliary excretion of copper, excessive copper accumulation mainly in the liver and basal ganglia can lead to both hepatic dysfunction and extrapyramidal motor as well as psychiatric symptoms. Moreover, cognitive dysfunction and additional depressive symptoms are frequently reported in these patients [23]. This suggests that MDD patients may similarly exhibit copper overload in the liver and basal ganglia. Recent studies have also found that abnormal copper concentrations can induce cell death, disrupt synaptic function or neural metabolism, and contribute to memory impairment in depressed patients [24,25]. Recent investigations aim to elucidate the predictive value of CRGs for MDD and their influence on the immunological microenvironment. In addition, a recent study identified hub genes associated with cuproptosis that contribute to the immune microenvironment in UC using bioinformatic analysis and experimental verification [26]. However, no studies have identified CRGs that simultaneously influence both IBD and MDD, nor have any drugs been identified that effectively target CRGs to modulate the progression of both IBD and MDD.

To gain a deeper understanding of the regulatory mechanisms of CRGs in patients with MDD and IBD, and to identify drugs that can effectively treat both conditions, we employed bioinformatics approaches to explore the shared regulatory mechanisms of cuproptosis in these two diseases. In this study, we identified five DE-CRGs that may serve as potential drug targets, as well as five key CRGs associated with both IBD and MDD, providing new therapeutic targets and prospects for the treatment of these diseases. Additionally, we predicted several drugs with the potential to treat both IBD and MDD by targeting DE-CRGs. These findings may contribute to further studies on the pathogenesis of IBD and MDD, as well as promote the development of novel therapeutic drugs for both diseases.

## 2. Materials and Methods

### 2.1. Data Source

The datasets were all from the GEO database, with samples from colon biopsies in both the CD and UC datasets, and the datasets used for CD were GSE20881 (including 73 control samples and 99 CD samples) [27], GSE24287 (including 25 control samples and 47 CD samples) [28], GSE179285 (including 31 control samples and 168 CD samples) [29]. The datasets used for UC were GSE13367 (including 20 control samples and 34 UC samples) [30], GSE24287 (including 25 control samples and 27 UC samples) [28], GSE179285 (including 31 control samples and 55 UC samples) [29]. MDD used two datasets GSE98793 for peripheral blood samples (including 64 control samples and 128 MDD samples) [31] and GSE19738 for whole blood samples (including 66 control samples and 66 MDD samples) [32]. MDD also selected datasets GSE54568 from the prefrontal cortex of the human brain, GSE54571 from the anterior cingulate cortex of the human brain, and GSE54564 from the amygdala tissue of the human brain. The GSE54568 dataset included 15 control samples and 15 MDD samples [33], the GSE54571 dataset included 13 control samples and 13 MDD samples [33], the GSE54564 dataset included 21 control samples and 21 MDD samples [33]. Then, 18 CRGs (*SLC25A3*, *SCO2*, *LIPT1*, *DLAT*, *PDHA1*, *MTF1*, *FDX1*, *NFE2L2*, *NLRP3*, *ATP7B*, *SLC31A1*, *LIAS*, *DLD*, *PDHB*, *GLS*, *CDKN2A*, *DBT*, *DLST*) were obtained from previous reports [34,35]. We combined the three datasets of CD, the three datasets of UC, and the two whole blood datasets of MDD separately. The batch effects among these datasets were eliminated by applying the combat algorithm in the “sva” R package [36] (v3.52.0).

### 2.2. Identification of Differentially Expressed Cuproptosis Genes

The differentially expressed CRGs were identified between normal and disease samples in CD, UC, and various tissues of MDD using the wilcox.test with a *p*-value < 0.05. The results of the differential analysis were then visually presented using box plots generated through the ggpubr package (v0.6.0).

### 2.3. Immune Cell Infiltration Analysis in CD, UC, and MDD

The CIBERSORT algorithm of the ggpubr package (v0.6.0) was utilized to analyze the infiltrations of 22 immune cells based on RNA expressions [37]. Differential immune cell infiltrations were identified between normal and disease samples with *p* < 0.05. Additionally, correlations between RNA expression of CRGs and infiltrations of the 22 immune cells were calculated.

### 2.4. Identification and Exploration of CD, UC, and MDD Subtypes

We used the “ConsensuClusterPlus” R package [38] (v1.68.0) to perform an unsupervised hierarchical cluster analysis of all disease samples (n = 673). The consensus matrix, consistent cluster score (>0.9), and cumulative distribution function (CDF) curve were evaluated to select the optimal number of clusters, which was K = 2, and the maximum number of clusters was K = 9. Next, the differential expressions of CRGs and infiltrations of immune cells were compared between subtypes. Then, we compared the differential RNA expressions and immune cell infiltrations between subgroups. Finally, gene set variation analysis (GSVA) was used to identify the different hallmark pathways between subgroups under unsupervised and parameter-free conditions [39].

### 2.5. Machine Learning Screening of Candidate Important Genes

In our study, we utilized four machine learning algorithms including Support Vector Machine (SVM) [40], Random Forest (RF) [41], Generalized Linear Model (GLM) [42], and eXtreme Gradient Boosting (XGB) [43] to predict the importance of CRGs in distinguishing normal and disease samples. The top 8 genes of each machine learning algorithm were selected as important genes, among which those considered as important genes by all three machine learning algorithms were deemed to be crucial factors leading to the occurrence of the disease.

### 2.6. Drug Predict

The oncoPredict (v 0.2) package was used to predict the IC50 values of each drug from The Cancer Therapeutics Response Portal (CTRP) database based on the differential CRG expressions between normal and disease samples in IBD and MDD [44]. We calculated the correlations between the IC50 values of potential drugs and RNA expressions of different CRGs.

### 2.7. Molecular Docking

The protein structure retrieved from the Protein Data Bank (https://www.uniprot.org/ accessed on 1 November 2024) was utilized as the model and the ligands were removed. Structures of drugs were found from ZINC15 (http://zinc15.docking.org/ accessed on 1 November 2024) and the Pubchem (https://pubchem.ncbi.nlm.nih.gov/ accessed on 1 November 2024) Small Molecule Ligand Database. We used PyMol software (v3.0.5) to conduct molecular docking and visualization of protein receptors with drugs and calculate the binding free energy.

### 2.8. Molecular Dynamics Simulations

Molecular dynamics (MD) simulations are widely used to understand the stability and dynamic behavior of proteins and protein–ligand complexes [45,46,47,48]. In this study, we utilized the GROMACS (v2023) MD simulation package to perform the MD simulation of proteins and protein–ligand complexes [49]. The topology and parameters for the protein were generated using the AMBER99SB force field, while the ligand topology and parameters were generated using the Chimera server [50]. Various structural parameters were measured, including root-mean-square deviation (RMSD), radius of gyration (Rg), solvent-accessible surface area (SASA), root-mean-square fluctuation (RMSF), hydrogen bonds, secondary structure, angles, and distances. The trajectories were visualized using CMD (v10.0.26100.2894) and the graphs were prepared using DuIvyTools software (v0.5.0) (https://duivytools.readthedocs.io/ accessed on 20 November 2024).

### 2.9. Phenome-Wide Association Studies

To delve deeper into side effects associated with 18 CRGs, we carried out phenome-wide association studies (PheWAS) for a wide range of diseases. We carried out an analysis of the impacts of SNPs on outcomes using summary statistics. The dataset we used came from the UK Biobank cohort, with a sample size reaching as high as 408,961 individuals [51]. For GWAS, we utilized SAIGE (v.0.29), a generalized mixed model method, to make adjustments for the uneven distributions of cases and controls. We defined the phenotypic outcomes of diseases in accordance with “PheCodes”. It is a system that categorizes codes from the International Classification of Diseases and Related Health Problems (ICD-9/-10), where these codes are linked to specific phenotypic manifestations [52]. This standardization allowed for more precise exploration of genetic–phenotypic associations.

### 2.10. Statistical Analysis

The vast majority of studies used R (v4.4.1). In all statistical analyses, *p* < 0.05 was significantly different.

## 3. Results

### 3.1. Screening of DEGs for IBD and MDD

To investigate whether cuproptosis could serve as a common regulatory mechanism in the development of IBD and MDD, we sought to identify differentially expressed CRGs that may be closely associated with the onset and progression of these conditions. To achieve this, we retrieved gene expression data from the GEO database, encompassing samples from CD colon, UC colon, MDD whole blood, MDD prefrontal cortex, MDD anterior cingulate cortex, and MDD amygdala for differential expression analysis. Subsequently, we analyzed the differentially expressed genes between normal and disease samples in CD, UC, and MDD, respectively. In total, we identified 11 DE-CRGs in CD, including 3 upregulated genes (*ATP7B*, *SLC31A1*, *DLST*) and 8 downregulated genes (*SLC25A3*, *DLAT*, *PDHA1*, *LIAS*, *DLD*, *PDHB*, *GLS*, *DBT*) (Figure 1A). In UC, we detected 11 DE-CRGs, comprising 1 upregulated gene (*NLRP3*) and 10 downregulated genes (*SLC25A3*, *LIPT1*, *PDHA1*, *FDX1*, *NFE2L2*, *LIAS*, *DLD*, *GLS*, *CDKN2A*, *DBT*) (Figure 1B). In MDD whole blood, we observed two DE-CRGs: one upregulated gene (*NFE2L2*) and one downregulated gene (*DBT*) (Figure 1C). In the MDD prefrontal cortex, two downregulated DE-CRGs were identified (*DLD*, *PDHB*) (Figure 1D). In the MDD anterior cingulate cortex, one downregulated DE-CRGs (NLRP3) was detected (Figure 1E). Finally, in the MDD amygdala, we identified two DE-CRGs: one upregulated gene (DLST) and one downregulated gene (DLAT) (Figure 1F). Simultaneously, the heatmap illustrated the expression profiles of CRGs in the CD colon samples, UC colon samples, MDD whole blood samples, MDD prefrontal cortex samples, MDD anterior cingulate cortex samples, MDD amygdala samples, and corresponding control samples (Appendix A). Additionally, an overview of the CRG expression was visualized using the Circos plot (Figure 1G). Seven DE-CRGs were identified at the intersection of IBD and MDD using a Venn diagram (Figure 1H), including the following: *DLD* (downregulated in both IBD and MDD), *DLAT* (downregulated in both IBD and MDD), *DLST* (upregulated in both IBD and MDD), *NLRP3* (upregulated in IBD, downregulated in MDD), *PDHB* (downregulated in both IBD and MDD), *DBT* (downregulated in both IBD and MDD), and *NFE2L2* (downregulated in IBD, upregulated in MDD). The findings suggest that CRGs exhibit both similar and distinct regulatory mechanisms in IBD and MDD.

### 3.2. Correlation Analysis of the 18 CRGs

To accurately analyze the potential linkage network of CRGs in both the control and disease groups, we conducted detailed correlation analyses of CRGs in each group. Figure 2A–F illustrate the correlations between CRGs in the disease and control groups, respectively. The lower-left corner represents the control group, while the upper-right corner represents the disease group. Significant differences in CRG correlations were observed between the disease and the normal groups. For example, the correlation among 18 CRGs was significantly higher in the disease group compared to the control group in the prefrontal cortex samples of the MDD brains. Conversely, in the anterior cingulate cortex samples, the correlation was significantly lower in the disease group than in the control group. These analyses disclosed significant correlations associated with CRGs under distinct conditions, revealing potential interrelationships of these genes. The findings provide valuable data support and theoretical insights for further exploration of the key mechanisms underlying disease progression.

### 3.3. Evaluation of Immune Cell Infiltration

We performed immune cell infiltration analysis in patients with IBD and MDD based on RNA expression data from the colon samples of CD and UC patients, as well as the whole blood, prefrontal cortex, anterior cingulate cortex, and amygdala samples from MDD patients (Figure 3A–F). The results showed that naive B cells, γ delta T cells, and M2 macrophages were significantly reduced in the IBD and MDD samples compared to normal subjects, whereas resting NK cells, monocytes, and M1 macrophages were significantly enriched. Plasma cells were increased in the CD samples and MDD whole blood samples but decreased in the MDD amygdala samples. Meanwhile, we analyzed the correlations between 18 CRGs and immune cell infiltration in the CD colon samples, UC colon samples, MDD whole blood samples, MDD prefrontal cortex samples, MDD anterior cingulate cortex samples, and MDD amygdala samples (Figure 4A–F). The findings indicated the following correlations: *CDKN2A* and *SLC31A1* were positively correlated with resting NK cells. *NLRP3* and *SCO2* were positively correlated with monocytes. *LIPT1*, *NLRP3*, *SLC31A1*, and *GLS* were positively correlated with M1 macrophages. *SLC25A3*, *PDHB*, *PDHA1*, *LIAS*, *DLD*, *DLAT*, and *DBT* were negatively correlated with resting NK cells. *SLC25A3*, *PDHA1*, *LIAS*, *GLS*, *DLD*, *DBT*, *DLAT*, and *CDKN2A* were negatively associated with M1 macrophages. These results suggest that CRGs may influence the development of IBD and MDD by modulating the infiltration of relevant immune cells through positive or negative regulation.

### 3.4. Subtype Analysis with CRGs

According to the expression profiles of CRGs, we integrated samples from CD, UC, and MDD for a comprehensive cluster analysis. Using the consensus matrix diagram (Figure 5A), consensus clustering score (Appendix A), cumulative distribution function (CDF) curve (Appendix A), and relative change in the area under the CDF curve (Appendix A), we determined that k = 2 was the optimal clustering number. Consequently, we identified two subtypes associated with CRGs: C1 (n = 430) and C2 (n = 243). Subtype C1 comprised 314 CD samples and 116 UC samples, while C2 included 194 MDD whole blood samples, 15 MDD prefrontal cortex samples, 13 MDD precingulate cortex samples, and 21 MDD amygdala samples (Figure 5B). PCA (Figure 5C) revealed a clear distinction between the two clusters. Additionally, the box plot (Figure 5D) demonstrated that the expression levels of *SLC25A3*, *SCO2*, *LIPT1*, *DLAT*, *PDHA1*, *MTF1*, *FDX1*, *NFE2L2*, *NLRP3*, *ATP7B*, *SLC31A1*, *LIAS*, *DLD*, *PDHB*, *GLS*, *CDKN2A*, *DBT*, and *DLST* were significantly higher in C2 compared to C1. The significant differences in CRG expression highlighted the distinct regulatory mechanisms governing cuproptosis in IBD and MDD.

Furthermore, immune cell infiltration analysis showed that, except memory B cells and M0 macrophages, other immune cell types exhibited significant differences between the two clusters (Figure 5E). GSVA analysis (Figure 5F) indicated that cluster 2 was mainly enriched in pathways related to peroxisome, TGF-β signaling, adipogenesis, hedgehog signaling, unfolded protein response, and protein secretion, whereas cluster 1 was mainly enriched in pathways associated with KRAS signaling up, angiogenesis, apical junction, glycolysis, pancreas β cells, and cholesterol homeostasis. These analyses provide a comprehensive and detailed investigation into the distinct regulatory mechanisms governing cuproptosis in IBD and MDD.

### 3.5. Machine Learning Screening for the Important Genes

Due to the limited sample size of the prefrontal cortex, anterior cingulate cortex, and amygdala samples from MDD patients, the accuracy of the machine learning model was compromised. Consequently, we selected colon tissue samples from CD and UC patients, as well as whole blood samples from MDD patients, for this analysis. We employed SVM, RF, XGB, and GLM algorithms to identify the top eight genes for CD (Figure 6A), UC (Figure 6B), and MDD whole blood (Figure 6C) patients, respectively. Using a Venn diagram, we identified genes commonly shared by three or more machine learning algorithms for CD (Figure 6D), UC (Figure 6E), and MDD whole blood (Figure 6F) patients, respectively.

The top eight genes for CD include SLC31A1, DLST, GLS, LIAS, DLD, and SLC25A3. For UC, the top eight genes are PDHA1, DLD, SCO2, DLAT, LIAS, CDKN2A, and GLS. In the case of MDD, the top eight genes are LIAS, SCO2, DLD, CDKN2A, SLC31A1, DBT, and NFE2L2. Consequently, DLD, LIAS, SCO2, CDKN2A, and SLC31A1 were identified as important factors implicated in the pathogenesis of both IBD and MDD. This discovery not only enhances our understanding of the molecular commonalities underlying these two seemingly distinct diseases but also paves the way for further exploration of potential therapeutic targets and diagnostic biomarkers.

### 3.6. Drug Prediction and Molecular Docking Analysis

Based on the expression levels of DE-CRGs associated with IBD and MDD, we identified seven potential therapeutic drugs for CD. These drugs positively regulated the downregulated CRGs and negatively regulated the upregulated CRGs in the CD samples (Figure 7A). Additionally, we identified 317 drugs for UC, 70 drugs for MDD whole blood, 114 drugs for the MDD prefrontal cortex, 2 drugs for the MDD anterior cingulate cortex, and 134 drugs for the MDD amygdala samples, respectively (Appendix A). Notably, we also identified 37 drugs with the potential to treat both IBD and MDD (Appendix A).

Based on the docking scores between these 37 drugs and proteins of five DE-CRGs (Figure 1H) with similar regulatory trends in IBD and MDD (Appendix A), we selected the drug with the lowest docking score for each gene protein to demonstrate the binding modes of proteins and drugs (Figure 7B–F). Specifically, barasertib interacted with DLAT, forming four hydrogen bonds with the L321, K250, and N251 amino acids (Figure 7B). Neratinib interacted with DLD, forming two hydrogen bonds with the Thr-44 and Val-357 amino acids (Figure 7C). Birinapant interacted with DLST, forming three hydrogen bonds with the Arg-170 and Glu-173 amino acids (Figure 7D). Neratinib interacted with PDHB, forming one hydrogen bond with the Pro-268 amino acids (Figure 7E). Birinapant interacted with DBT, forming five hydrogen bonds with the Thr-51, Ser-10, and Asp-85 amino acids (Figure 7F).

According to the lowest docking score table presented in Appendix A, the binding energy between DLAT and barasertib exhibited the lowest value of −46.317345. This finding strongly indicates that DLAT and barasertib have the most potent binding affinity among the studied pairs, suggesting a potentially significant interaction with important implications for the underlying biological mechanisms being investigated.

### 3.7. Molecular Dynamics Simulation

To further characterize the interactions between proteins of DE-CRGs and the potential therapeutic drugs, we selected DLAT for molecular dynamics simulation due to its strongest binding affinity with multiple candidate drugs. We compared the interactions of DLAT with the top four drugs exhibiting the lowest docking scores: barasertib, neratinib, alisertib, and NVP-TAE684. Alisertib was excluded from further analysis due to possible uncertainties in its chemical structure and its suboptimal binding affinity to protein macromolecules.

In molecular dynamics simulations, the RMSD (root-mean-square deviation) plot was used to quantify the structural deviations between the molecular structure and the reference structure over the course of the simulation (Figure 8A). This metric provides insight into the extent of structural changes occurring over time. Low RMSD values of barasertib and NVP-TAE684 indicate that these molecules maintained stable conformations close to the reference structure throughout the simulation, whereas the higher RMSD value of neratinib suggests significant structural deviation from the reference structure.

We subsequently utilized the Rg (radius of gyration) plot to quantify the extent of molecular expansion during the simulation (Figure 8B). The low Rg values for barasertib and NFP-TAe684 indicate that these molecules remained more compact, whereas the higher Rg values for neratinib suggest a more expanded conformation during the simulation. Meanwhile, the RMSF (root-mean-square fluctuation) plot was used to quantify the volatility of each atom within the molecule (Figure 8C). Lower RMSF values indicate tighter packing of the protein and small-molecule drug system. We observed that the DLAT–barasertib system exhibited tighter packing compared to the other two systems (Figure 8C). Finally, the SASA (solvent-accessible surface area) plot was employed to assess the extent of molecular surface exposure to solvent molecules during the simulation (Figure 8D). The observed decrease in SASA values for all three drugs suggests that their molecular surfaces became more concealed or reduced, indicating a more compact or tightly folded internal structure, thereby reducing solvent contact. Based on the comprehensive molecular dynamics simulations with DLAT as the drug target, barasertib and NTP-TAE684 have been identified as the most promising lead compounds for the treatment of IBD and MDD, offering new perspectives for therapeutic interventions in these complex and debilitating disorders.

### 3.8. Analysis of Phenome-Wide Association Studies of 18 CRGs

To investigate the phenotypes associated with 18 CRGs, we conducted a PheWAS analysis to examine the potential links between proteins of 18 CRGs and 1403 diseases and traits in the UK Biobank. After screening, it was found that DLAT, DLST, and PDHA1 had no eligible SNPs in the GWAS database. Following FDR correction, the PheWAS analysis revealed that DLD (both as a co-DEG and a co-DIG) was significantly correlated with two categories of traits (PFDR < 0.05): endocrine diseases (type 2 diabetes and diabetes mellitus) and injuries (upper limb fractures). In contrast, LIAS did not exhibit any significant trait association (Figure 9). Additionally, the following genes were linked to specific traits (PFDR < 0.05, Appendix A): MTF1: acute tonsillitis and secondary malignancy of respiratory organs. ATP7B: tinnitus and viral hepatitis. CDKN2A: alcoholism, liver damage, and breast cancer. DBT: subarachnoid hemorrhage and infection/inflammation of internal prosthetic devices, implants, and grafts. FDX1: bacterial enteritis and intestinal infections. GLS: benign neoplasm of the uterus and uterine leiomyoma. LIPT1: schizophrenia and other psychotic disorders, chronic pharyngitis, and nasopharyngitis. NLRP3: benign neoplasms of unspecified sites and septicemia. PDHB: leukemia and streptococcus infection. NFE2L2: coagulation defects and intracerebral hemorrhage. SCO2: psychological and somatoform disorders, and ectopic/contact dermatitis from unspecified causes. SLC25A3: schizophrenia and peptic ulcer (excluding esophageal). SLC31A1: inflammatory and toxic neuropathy, and other disorders of the urethra and urinary tract. Evidently, CRGs are associated with the aforementioned phenotypes. Notably, when FDX1, LIPT1, and SLC25A3 are considered as potential drug targets, there is a concerning likelihood that they may precipitate an accelerated onset of colitis and depression. Therefore, these potential risks warrant cautious consideration and further in-depth investigation to mitigate such adverse outcomes.

## 4. Discussion

Previous observational studies have suggested a bidirectional association between IBD and MDD [53]. However, the potential mechanisms underlying this relationship remain to be fully elucidated. Cuproptosis, an independent form of cell death, has been shown to be highly correlated with mitochondrial respiration and the lipoic acid (LA) pathway [54]. Cuproptosis occurs through copper binding to lipoylated enzymes in the tricarboxylic acid (TCA) cycle, leading to subsequent protein aggregation, proteotoxic stress, and ultimately cell death [55]. Recent studies have implicated cuproptosis in the progression of both IBD and MDD. However, the expression patterns and potential regulatory mechanisms of CRGs have not yet been compared between these two diseases. This study aims to explore the potential common regulatory mechanisms of CRGs in the development of both IBD and MDD, as well as the potential roles of these genes in influencing the co-occurrence of the two conditions. We identified five common DE-CRGs in IBD and MDD: *DLD*, *DLAT*, *DLST*, *PDHB*, and *DBT*. We then predicted drugs that may affect the expression of DE-CRGs, resulting in a list of 37 potential drugs. Subsequently, we performed molecular docking analyses on the five common DE-CRGs and the 37 drugs. Given that DLAT exhibited the lowest binding energy with the drugs, we selected the top three drugs with the lowest binding energies for molecular dynamics simulations with DLAT. This process identified barasertib and NTP-TAE684 as the two most effective compounds. Furthermore, using four machine learning algorithms, we identified five significant CRGs associated with both IBD and MDD: *DLD*, *LIAS*, *SLC31A1*, *SCO2*, and *CDKN2A*. Additionally, we investigated phenotypes related to CRGs and the results suggest that FDX1, LIPT1, and SLC25A3 may contribute to the progression of both IBD and MDD.

In this study, we initially analyzed the expression patterns of CRGs and identified five shared DE-CRGs in both IBD and MDD. These five common DE-CRGs have the potential to serve as drug targets for disease treatment. The shared alterations in the levels of these DE-CRGs across both diseases indicated that they may share common regulatory mechanisms, contributing to disease onset. Correlation analyses of CRGs showed potential links between genes in the control and disease groups. Subsequent cell infiltration analysis revealed that differentially expressed CRGs were associated with the infiltration of immune cells at inflammatory sites, impacting the abundance and spatial distribution of various immune cell subpopulations, as well as their dynamic correlation with the development of inflammation. These findings not only deepen our understanding of the intrinsic immunopathological mechanisms of inflammation but also provide important clues and directions for the development of novel anti-inflammatory therapies and precise inflammation diagnostic markers. This is expected to promote more targeted and effective clinical treatment of inflammation-related diseases. Subsequently, the results of an immune cell analysis demonstrated that the proportions of naive B cells, γ delta T cells, and M2 macrophages were significantly lower in the IBD and MDD samples compared to normal subjects. In contrast, resting NK cells, monocytes, and M1 macrophages were significantly enriched in the IBD and MDD samples. Notably, plasma cell levels were elevated in the CD samples and whole blood samples from patients with MDD but decreased in the MDD amygdala samples. These results suggest that CRGs may influence the development of IBD and MDD by positively or negatively regulating the infiltration of immune cells.

To understand the correlations of CRGs with all samples from both diseases, we divided all samples into two subtypes, colitis subtype C1 and depression subtype C2. We then performed PCA, immune cell, and GSVA analyses. These analyses revealed significant differences between the IBD and MDD samples, suggesting that the physiological processes involving these genes are more active in MDD. Subsequently, we performed machine learning algorithms, including SVM, RF, XGB, and GLM, to identify common significant CRGs in both diseases. The identified CRGs include *DLD*, *LIAS*, *SCO2*, *CDKN2A*, and *SLC31A1*. Notably, DLD, also known as dihydrolipoamide dehydrogenase, functions as a common differentially expressed and significantly associated CRG. It is a key component of the glycine cleavage system and serves as an E3 component of the three α-keto acid dehydrogenase complexes.

Based on the changes in common DE-CRGs observed in the diseases, we screened drugs for the simultaneous treatment of CD, UC, and MDD through drug sensitivity analysis. Drugs with the lowest binding energy to each target gene were selected for molecular docking. DLAT and barasertib exhibited the lowest binding energy scores and then molecular dynamics simulations were conducted on the top three drugs with the lowest free energies targeting DLAT, revealing that barasertib and NTP-TAE684 had the most promising therapeutic effects. Given that the potential physiological effects of barasertib and NTP-TAE684 in IBD and MDD have not yet been thoroughly studied, it is imperative to establish an animal model exhibiting symptoms of both diseases to evaluate whether these two drugs can ameliorate both conditions in future studies. Considering the potential side effects associated with targeting CRGs, prior to application of the predicted drugs targeting CRGs for the treatment of IBD and MDD, the possible risks of physiological toxicities to multiple organs should be carefully evaluated in animal models of both diseases during the treatment process.

Excessive copper induces cuproptosis by promoting protein toxic stress reactions via copper-dependent anomalous oligomerization of lipoylation proteins in the tricarboxylic acid (TCA) cycle and reducing iron–sulfur cluster protein levels [56]. Based on the correlation analysis, we found that *FDX1* was positively associated with *DLAT* in the disease group and with *LIAS* in the control group. Ferredoxin1 (FDX1) promotes dihydrolipoyl transacetylase (DLAT) lipoacylation and reduces iron–sulfur cluster proteins by converting Cu^2+^ to Cu, thereby inducing cell death [56]. FDX1 directly regulates protein lipoylation by binding the lipoyl synthase (LIAS) enzyme, enhancing its functional interaction with the lipoyl carrier protein GCSH, rather than indirectly regulating cellular Fe-S cluster biosynthesis [57]. Recent evidence has conferred a new role for TCA cycle intermediates, which are generally considered important for biosynthetic purposes, as signaling molecules that control chromatin modifications, DNA methylation, the hypoxic response, and immunity [58]. While numerous studies have demonstrated the close relationship between cuproptosis and IBD, relatively few articles have explored the connection between cuproptosis and MDD.

The high degree of heterogeneity and multifactorial nature of MDD renders existing pathological mechanisms inadequate to fully explain the condition, thereby increasing the difficulty in identifying effective drug targets. Recent studies have indicated that copper ions may be a key trace element in the neurobiological mechanisms of depression, as they are involved in various aspects of neurobiology and the biochemical mechanisms of antidepressants [24,59]. Elevated copper levels have been particularly associated with cognitive decline and anhedonia [24]. Although copper is essential for sustaining vital bodily functions, excessive accumulation within cells can lead to cell death, potentially contributing to the pathogenesis of depression [54]. The gastrointestinal epithelium serves as an absorptive barrier for a variety of nutrients, including metal ions, which regulate both microbial virulence and host immune responses [60]. Despite its importance, modern science has yet to establish clinical guidelines for micronutrition intake in IBD patients, considering factors such as trace element abundances, their balance, and individualized patient’s needs [61]. Copper metabolism plays an important role in gastrointestinal disorders. Excessive copper storage can cause intestinal cell damage and may lead to pathological conditions [62].

The UKB is a comprehensive biomedical database encompassing population health and genetic research resources, with over 500,000 participants aged 37 to 73 years recruited from 22 assessment centers. In this study, traits with a sample size of fewer than 1000 for dichotomous variables were excluded due to potential biases associated with the small sample size [63]. We conducted a full-phenotypic association analysis of genes related to cuproptosis. CRGs have been linked to various conditions. Our results indicated that the cuproptosis genes were associated with colitis and depression, while the mechanisms by which the CRGs affect the progression of both conditions need to be extensively studied in future.

The Mediterranean diet is widely regarded as one of the healthiest dietary patterns globally, attributed to its rich content of antioxidants and anti-inflammatory nutrients [64]. Extensive research has established a robust association between adherence to the Mediterranean diet and reduced incidence of certain chronic gastrointestinal diseases [65]. Given its protective effects against oxidative and inflammatory cellular processes, the Mediterranean diet is considered an effective and manageable strategy for reducing cancer incidence [64]. This diet is abundant in antioxidants such as vitamins C and E, and flavonoids, which neutralize free radicals and mitigate oxidative stress. In the context of cuproptosis, copper ions induce ROS, leading to oxidative stress. Antioxidants may counteract this effect, thereby reducing cellular sensitivity to copper ion toxicity and influencing cuproptosis. The polyunsaturated fatty acids present in this diet help maintain cell membrane fluidity and integrity, regulating copper ion transport and distribution, thus lowering the risk of cuproptosis. Additionally, the high fiber content promotes beneficial gut microbiota, which can influence host metabolism, immunity, and potentially regulate copper ion metabolism and excretion, affecting cuproptosis-related processes. Since it has been confirmed that cuproptosis is closely linked to IBD and MDD, the Mediterranean diet may potentially influence these conditions by modulating cuproptosis.

Nutritional care is a cornerstone of treatment of many diseases, with the potential to significantly improve patient outcomes by addressing malnutrition and enhancing recovery [66]. Based on the type of diseases, treatment approach, and individual circumstances, nutritionists develop personalized nutritional support plans to ensure adequate nutrient intake. By maintaining optimal nutritional status, patients can better tolerate surgery, reduce the risk of complications such as infections and anemia, alleviate symptoms like loss of appetite, and facilitate the smooth progress of treatment. In response to metabolic disorders caused by IBD, nutritionists adjust the dietary structure and provide specific nutrients to enhance immune function [67]. Copper ions can act as a double-edged sword in cells [68]. Nutritional care requires formulating dietary plans based on individual conditions, emphasizing the reasonable consumption of copper-containing foods. Excessive copper intake exceeding the body’s metabolic capacity may lead to copper ion accumulation in cells and induce cuproptosis. Conversely, insufficient copper intake can affect normal physiological functions.

This study has several limitations. Firstly, the sample size for MDD is relatively small. While it reflects certain features of the disease to some extent, the limited sample size restricts the generalizability and robustness of the findings. Future studies should expand the sample size to enhance the reliability and validity of the conclusions. Secondly, the absence of SNPs for DLAT, DLST, and PDHA1 in the GWAS database may lead to the overlooking of certain rare signs or comorbidities, potentially affecting the comprehensiveness of the study’s conclusions. Thirdly, although this study identified five differentially expressed CRGs that are common across conditions, these results are preliminary predictions. They lack in vivo and ex vivo experimental validation, and their authenticity and biological relevance require further confirmation through rigorous experimental methods. Additionally, bioinformatics analysis has identified barasertib and NTP-TAE684 as potential therapeutic drugs for both IBD and MDD at the molecular level. However, their effects and potential mechanisms at the cellular and organismal levels have not yet been investigated. Therefore, it is imperative to perform a series of comprehensive functional experiments, both in vitro and in vivo, to elucidate the underlying mechanisms linking drug action to biological phenotypes. This will enhance the reliability of our findings and provide valuable insights for future research in this field.

## 5. Conclusions

Our study identified five common DE-CRGs (*DLD*, *DLAT*, *DLST*, *PDHB*, and *DBT*) and five common disease-associated CRGs (*DLD*, *LIAS*, *SLC31A1*, *SCO2*, and *CDKN2A*) in both IBD and MDD. Meanwhile, our study predicted two potential therapeutic drugs, barasertib and NTP-TAE684, which may concurrently influence the progression of IBD and MDD. Future studies should focus on validating the expression and investigating the functional roles of the five DE-CRGs and the five disease-associated CRGs in both diseases. It is essential to evaluate the potential efficacy of the two drugs in both cellular and animal models of IBD and MDD. Such efforts may pave the way for future therapeutic breakthroughs in the treatment of both disorders. DE-CRGs identified in our study may elucidate the potential pathological connections between IBD and MDD, providing critical entry points for interdisciplinary research and expanding our understanding of mind–body interaction mechanisms. By utilizing five key CRGs, we can construct novel diagnostic marker combinations to enhance diagnostic accuracy. Early detection of IBD and MDD can be achieved by measuring the expression levels of specific genes before typical symptoms manifest, facilitating timely diagnosis and intervention. Furthermore, based on DE-CRGs, multi-target drugs can be developed to simultaneously address IBD- and MDD-related pathologies, thereby improving therapeutic outcomes.

## Figures and Tables

**Figure 1 genes-16-00339-f001:**
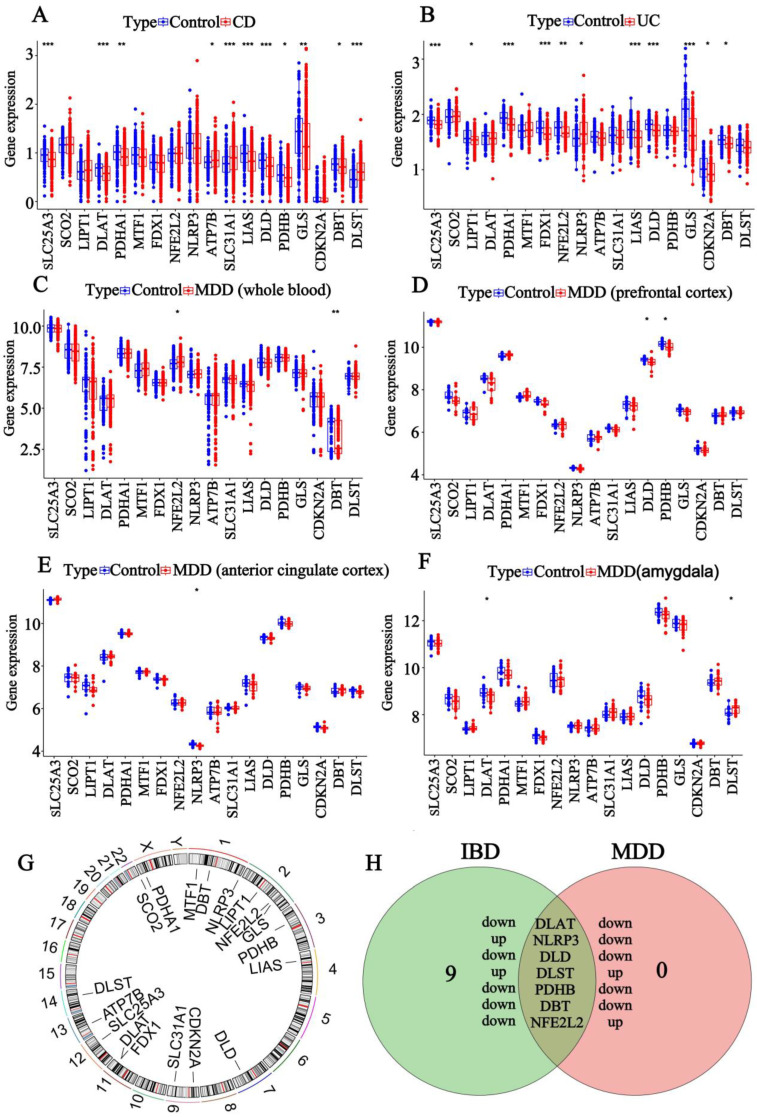
Identification of differentially expressed (DE-) CRGs associated with CD, UC, and MDD. (**A**). The box plot of DE-CRGs identified after merging GSE20881, GSE24287, and GSE179285 in CD. (**B**). The box plot of DE-CRGs identified after merging GSE13367, GSE24287, and GSE179285 in UC. (**C**). The box plot of DE-CRGs identified after merging GSE98793 and GSE19738 of MDD whole blood. (**D**). The box plot of DE-CRGs identified in GSE54568 of MDD prefrontal cortex. (**E**). The box plot of DE-CRGs identified in GSE54571 of MDD anterior cingulate cortex. (**F**). The box plot of DE-CRGs identified in GSE54564 of MDD amygdala. (**G**). Circos diagram shows the positions of CRGs in the chromosome. (**H**). Venn diagram shows the number of DE-CRGs and overlapping DE-CRGs among the conditions including IBD and MDD. (*: *p* < 0.05, **: *p* < 0.01, ***: *p* < 0.001).

**Figure 2 genes-16-00339-f002:**
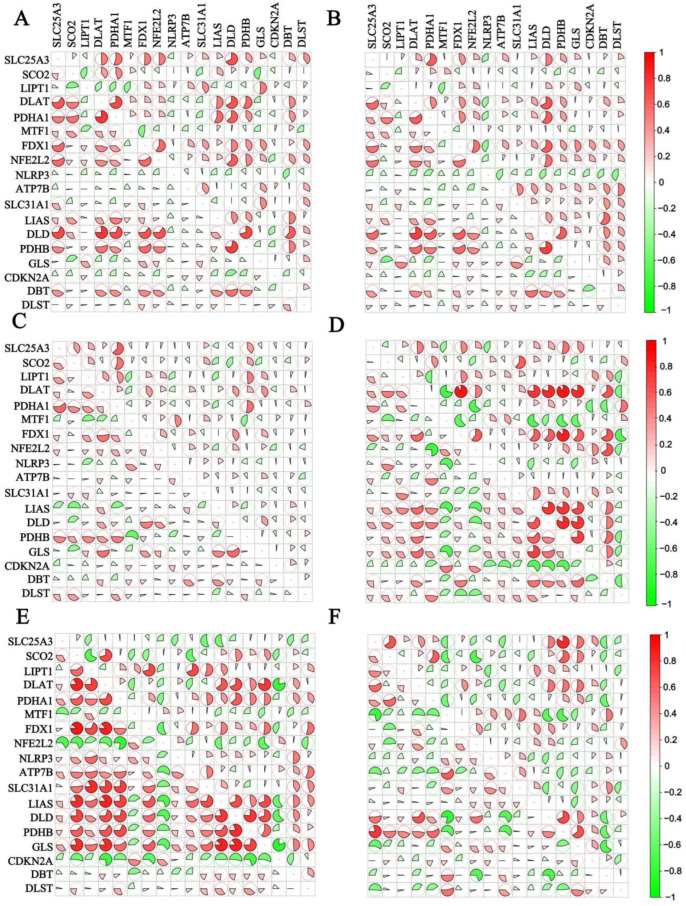
Correlation analysis of CRGs in control group and disease group. (**A**). Correlation analysis of CRGs in normal and disease groups after merging GSE20881, GSE24287, and GSE179285 in CD. (**B**). Correlation analysis of CRGs in normal and disease groups after merging GSE13367, GSE24287, and GSE179285 in UC. (**C**). Correlation analysis of CRGs in normal and disease groups after merging GSE98793 and GSE19738 of MDD whole blood. (**D**). Correlation analysis of CRGs in normal and disease groups in GSE54571 of MDD prefrontal cortex. (**E**). Correlation analysis of CRGs in normal and disease groups in GSE54571 of MDD anterior cingulate cortex. (**F**). Correlation analysis of CRGs in normal and disease groups in GSE54564 of MDD amygdala.

**Figure 3 genes-16-00339-f003:**
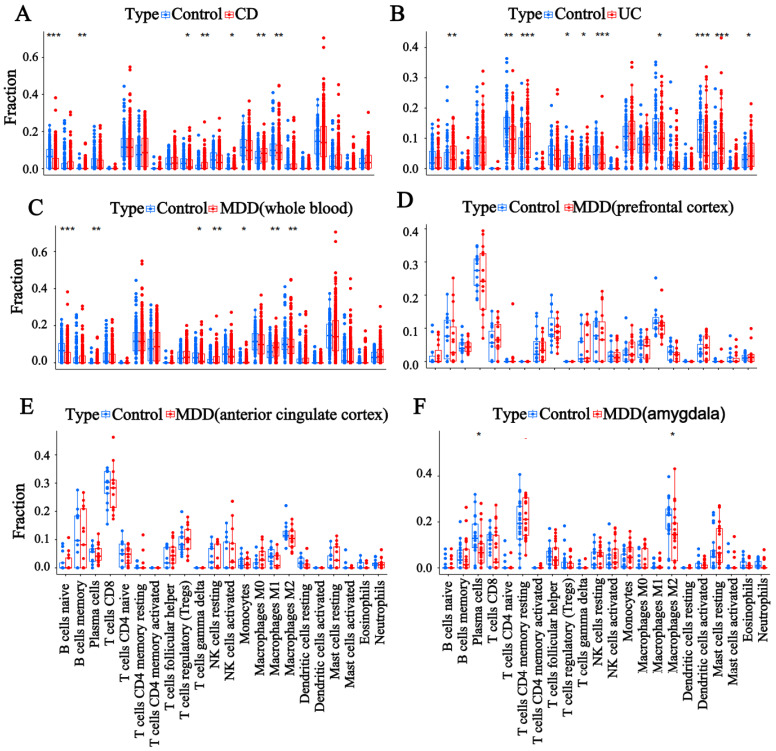
Analysis of the immune cell microenvironment. Box plot shows the differences in immune cell infiltration between (**A**) CD colon samples, (**B**) UC colon samples, (**C**) MDD whole blood samples, (**D**) MDD prefrontal cortex samples, (**E**) MDD anterior cingulate cortex samples, (**F**) MDD amygdala samples, and control samples. (*: *p* < 0.05, **: *p* < 0.01, ***: *p* < 0.001).

**Figure 4 genes-16-00339-f004:**
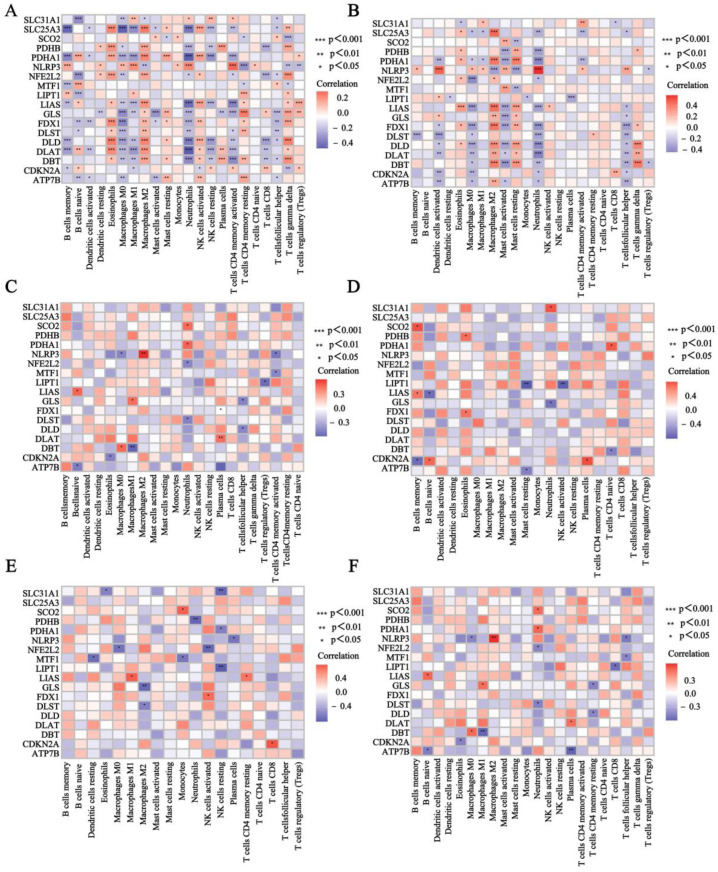
The correlations between 18 CRGs and immune cell infiltration. The heatmap shows the correlation of CRGs with immune cells in (**A**) CD colon samples, (**B**) UC colon samples, (**C**) MDD whole blood samples, (**D**) MDD prefrontal cortex samples, (**E**) MDD anterior cingulate cortex samples, (**F**) MDD amygdala samples.

**Figure 5 genes-16-00339-f005:**
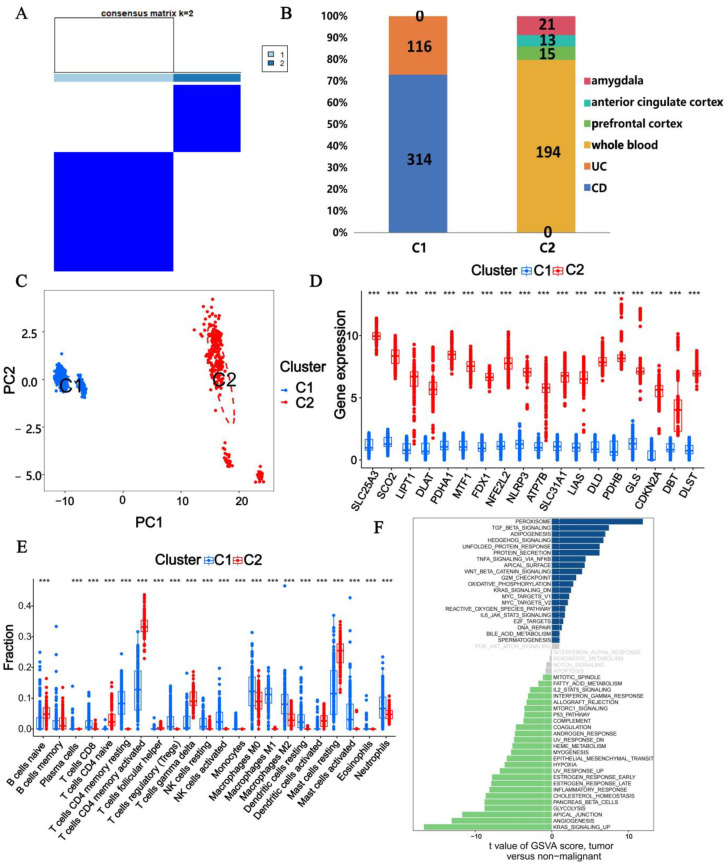
Identification of the molecular subtypes in IBD based on the expression of CRGs. (**A**) Consensus heatmaps when k  =  2. (**B**) Proportion of disease samples in subtypes. (**C**) PCA of two subtypes. (**D**) Box plot shows the expression of CRGs between two subtypes. (**E**) Box plot shows the differences in immune cell infiltration between two subtypes. (**F**) GSVA analysis between cluster 1 and cluster 2. (***: *p* < 0.001).

**Figure 6 genes-16-00339-f006:**
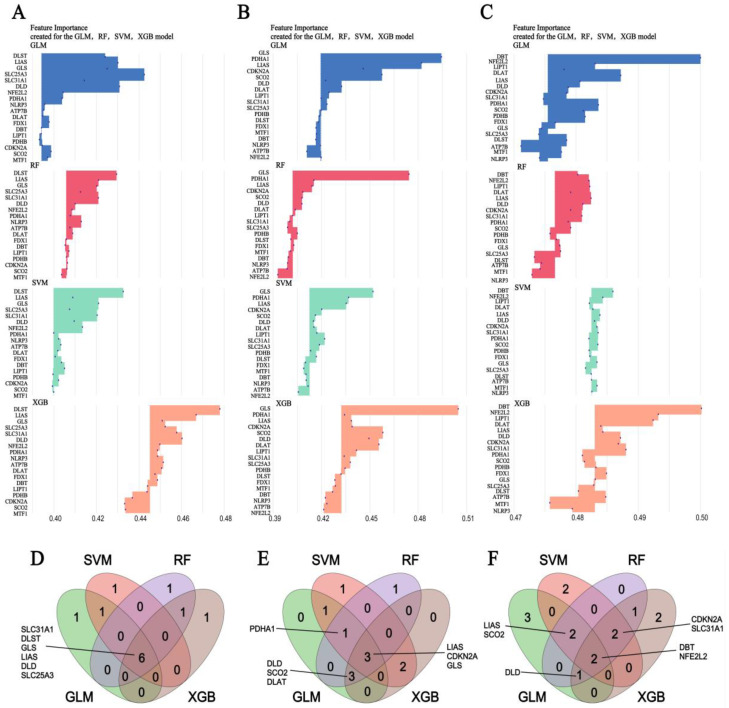
Identified important CRGs associated CD, UC, and MDD by machine learning algorithms. The top 8 genes associated with CD colon samples (**A**), UC colon samples (**B**), and MDD whole blood samples (**C**) identified by GLM, RF, SVM, and XGB. Wayne diagram of important genes shared by three or more machine learning algorithms in CD colon samples (**D**), UC colon samples (**E**), and MDD whole blood samples (**F**). · represent importance score.

**Figure 7 genes-16-00339-f007:**
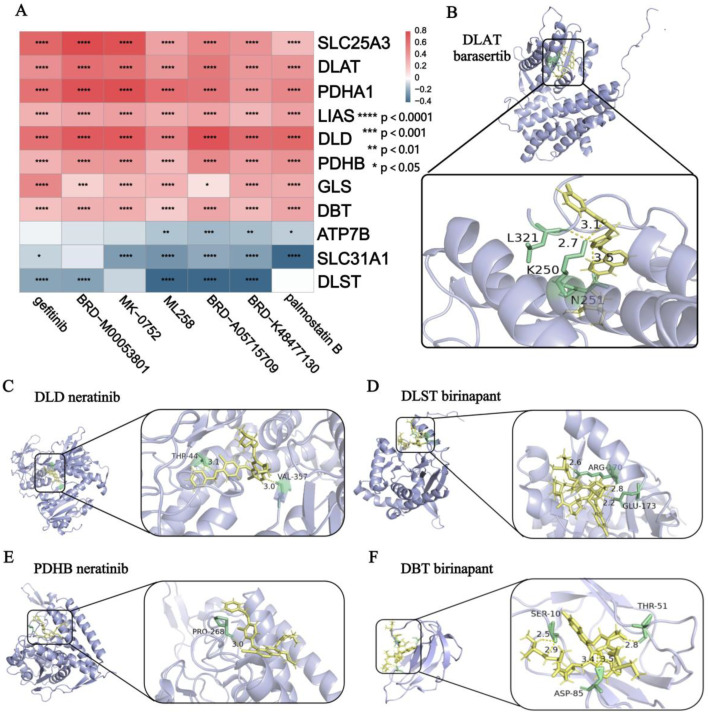
Drug predicted and molecular docking analysis. (**A**) Heatmap showing the correlations between drug sensitivity and expression level of DE-CRGs in CD. (**B**) DLAT bound with barasertib. (**C**) DLD bound with neratinib. (**D**) DLST bound with birinapant. (**E**) PDHB bound with neratinib. (**F**) BDT bound with birinapant.

**Figure 8 genes-16-00339-f008:**
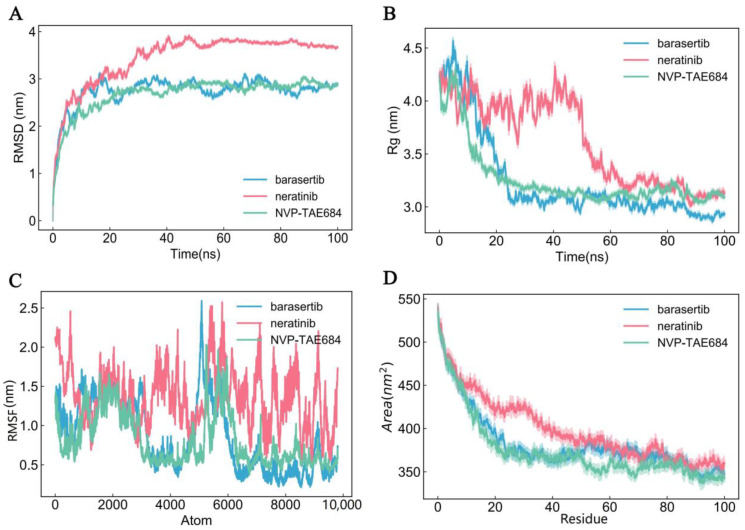
Molecular dynamics simulation analysis of DLAT with barasertib, neratinib, and NVP-TAE684. (**A**) RMSD, (**B**) Rg, (**C**) RMSF, (**D**) SASA experiments.

**Figure 9 genes-16-00339-f009:**
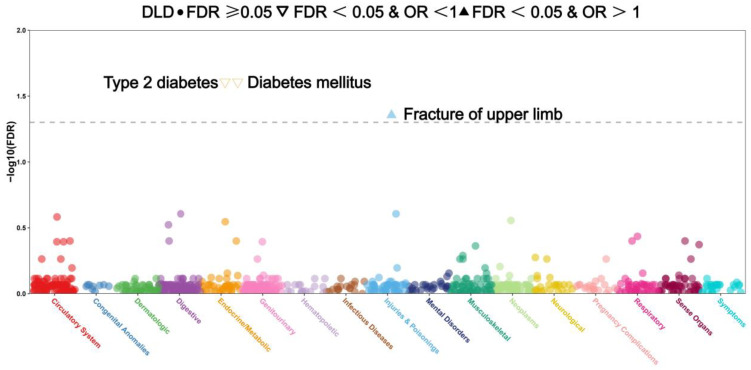
Phenome-wide association study (PheWAS) was conducted to meticulously dissect the associations between DLD and a diverse array of other disease outcomes, aiming to comprehensively explore and elucidate the potential broader implications and correlations of DLD within the spectrum of various diseases.

## Data Availability

Data of mRNA expression profiles were collected from GEO database (https://www.ncbi.nlm.nih.gov/geo/ accessed on 1 October 2024). Nine databases were included: GSE20881, GSE24287, GSE179285, GSE13367, GSE98793, GSE19738, GSE54568, GSE54571 and GSE54564. The full code used during the current study is available at https://github.com/sangmm12/Cuproptosis_IBD_MDD/ accessed on 27 February 2025.

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
