# Peer review of "Common Regulatory Mechanisms Mediated by Cuproptosis Genes in Inflammatory Bowel Disease and Major Depressive Disorder"

_genes, 2025, doi:10.3390/genes16030339_

Round 1
Reviewer 1 Report
Comments and Suggestions for Authors
This study identified genes associated with cuproptosis in relation to major depressive disorders using colonic CD/UC biopsies from the GEO database. The study is of interest and could be slightly improved with minor revisions.
Some minor suggestions:
- Specify in the introduction that anxious-depressive disorders can affect patients with IBD even in conditions of complete clinical remission (https://pubmed.ncbi.nlm.nih.gov/35346015/);
- Figure 1 A-F is poorly legible. It might be better to separate them from G and H and create two figures to enlarge the content; otherwise, it remains difficult to read;
- Increase the size of Figures 2, 3, and 5 slightly;
- Elaborate further on the translational potential of this study.
Reviewer 2 Report
Comments and Suggestions for Authors
The study proposed by the researchers for review should be read several times to understand the finer reasoning of our colleagues. The paper is absolutely unassailable for the data proposed for their analysis and for the statistics. But if we read the introduction, there is a fair mention of the intestinal microbiome, which we know is an expression of the microbiota that can be modified by the therapies that subjects affected by intestinal inflammatory diseases always take daily and by less varied foods compared to the population not affected by intestinal inflammatory disease. The microbiome, moreover, we have seen to be involved in other processes affecting the intestine such as neoplastic pathology (doi.org/10.3390/jcm13216578 to be read and cited in the bibliography) and even in post-anastomotic leakage (doi.org/10.3390/jcm13226634 to be read and cited in the bibliography). In light of these arguments we ask two questions to our colleagues, how important is the Mediterranean diet in the biochemical mechanisms so finely described? The second a nutritionist of the hospital team, often oncology, (DOI10.3390/nu17010188 to read and cite in bibliography) how important can it have in the follow-up of these patients? for the rest excellent iconography, excellent English, good bibliography
Comments on the Quality of English Languagegood english
